# Valorization of Historical Natural History Collections Through Digitization: The Algarium Vatova–Schiffner

**DOI:** 10.3390/plants13202901

**Published:** 2024-10-17

**Authors:** Linda Seggi, Raffaella Trabucco, Stefano Martellos

**Affiliations:** 1Department of Life Sciences, University of Trieste, 34127 Trieste, Italy; linda.seggi@phd.units.it; 2Fondazione Musei Civici di Venezia, Natural History Museum of Venice Giancarlo Ligabue, 30135 Venezia, Italy; raffaella.trabucco@fmcvenezia.it

**Keywords:** cultural heritage, Darwin Core, herbarium sheet, natural history museum, specimen

## Abstract

Digitization of Natural History Collections (NHCs) and mobilization of their data are pivotal for their study, preservation, and accessibility. Furthermore, thanks to digitization and mobilization, Natural History Museums can better showcase their collections, potentially attracting more visitors. However, the optimization of digitization workflows, especially when addressing small and/or historical NHCs, remains a challenge. Starting from a practical example, this contribution aims at providing a general guideline for the digitization of historical NHCs, with a particular focus on pre-digitization planning, during which some decisions should be made for ensuring a smooth, cost- and time-effective digitization process. The digitization of the algarium by Aristocle Vatova and Victor Schiffner was carried out following an *image-to-data* workflow, which allowed for reducing the handling of the specimens. The metadata were organized according to the Darwin Core standard scheme, and, together with the digital images of the specimens, have been made available to the scientific community and to the general public via an online portal. Thanks to the application of digital technologies and standardized methods, the accessibility of the collection has been enhanced, and its integration with historical data is possible, highlighting the relevance of shared experiences and protocols in advancing the digital transformation of natural history heritage.

## 1. Introduction

Natural History Collections (NHCs), which host specimens that have been collected by researchers, date back to the 17th century. While the first gatherings of natural history specimens together with objects of other natures are known to be the Renaissance cabinets of curiosities [1], modern NHCs appeared with the establishment of Natural History Museums (NHMs). The first NHM—as we define it today—was likely the National Museum of Natural History, Paris (1635), while the first NHM accessible to the general public was the Ashmolean Museum (1683) [2]. Given their long history, NHMs host a wealth of data—in the form of specimens and literature—that are pivotal for understanding the evolution of biodiversity in the Anthropocene [3], and for forecasting its evolution in the different possible scenarios related to global change. NHC specimens (and their metadata) are also relevant to a wide array of investigation topics, including taxonomy, ecology, history, social sciences, conservation, and benefit sharing [4,5]. NHCs are normally seen as a memory of natural heritage. However, they are also part of cultural heritage in its broader meaning, even if cultural and natural heritage are often regarded as mutually exclusive concepts [6].

Given the relevance of NHCs, the mobilization of specimen data by means of digitization has become pivotal [7]. Digitization, i.e., the creation of digital representations of specimens, is a means of increasing the usability and accessibility of NHCs [7,8,9,10]. While expensive, digitization can provide significant benefits in the medium and long terms [11], especially since it increases the fitness for use of NHCs across a broad array of disciplines. By enhancing the accessibility of specimen data under common standards and harnessing the power of distributed datasets, digitization can contribute to building new knowledge based on falsifiable data, and—at the same time—increasing natural history’s heritage value to society. Digital specimens have the potential to take NHCs to the next level [12,13], allowing for the creation of wider networks of extended specimens [14,15], in which specimen data are linked to their extensions, e.g., molecular, morphological and ecological data, distribution information, etc. This new global framework calls for the digitization of NHCs all over the world, prioritizing those which are the most relevant for research and/or society. High priority should be given as well to those collections that are at higher risk of loss, because of limited or poor curation, or simply because of natural deterioration over time.

This contribution aims at showcasing the digitization of a relevant historical collection of marine algae, the algarium of Aristocle Vatova and Victor Schiffner, which was carried out by the Natural History Museum of Venice Giancarlo Ligabue, Fondazione Musei Civici di Venezia, together with the Department of Life Sciences, University of Trieste. The project was developed and carried out as a guideline for the digitization of historical NHCs, aiming at facilitating digitization, curation, and data linking among NHMs. This contribution focuses especially on pre-digitization planning, during which poor decisions can influence the digitization process, causing it to lag or halt and/or leading to an increase in its costs.

## 2. Digitization of NHCs

The need for increasing global accessibility to NHC specimens has led to the evolution of novel digitization practices and to the development of massive, industrialized workflows for processing high volumes of specimens, mostly in botanical [16] and, more recently, in entomological NHCs. Such approaches are, however, not always feasible, especially in historical NHCs, where specimens are often preserved and/or mounted in different ways than in modern NHCs, or are too fragile (and, thus, more prone to deterioration), and so, more careful approaches are called for. Additionally, massive digitization can only be carried out if resource-intensive steps, such as specimen selection and databasing of associated information, are minimized [17]. Consequently, it is feasible only for certain large collections. As a result, NHC digitization workflows often rely on manual processes, and the speed at which digital images and metadata are produced and published is relatively limited [18].

Among massive, large-scale digitization projects, a recent example has been provided by “Digitize!” at the Museum für Naturkunde, Berlin [19]. Herbarium sheet mass-digitization workflows were showcased by De Smedt et al. [20] in the Meise Botanic Garden Herbarium, by Jardine et al. [21] in the NHM Herbarium, and by Haston et al. [22] in the Royal Botanic Garden Edinburgh Herbarium. As far as insects are concerned, several novel methods are being developed for mass digitization of specimens and labels [19,23,24,25,26]. Automated or semi-automated mass digitization workflows for microscope slides have been developed [27,28], while an interesting approach to mass digitization of wet-preserved specimens was showcased by Dupont et al. [29]. To our knowledge, digitization workflows on other NHCs, and, especially, on historical NHCs, are still being developed on a case-by-case basis, and can barely be replicated. Furthermore, while industrialized approaches are particularly effective in quickening the digital imaging process, pre- and post-digitization curatorial tasks, as well as the extraction of metadata from specimens’ labels, are mostly done manually. While curatorial steps can barely be automatized, the process of metadata extraction by means of artificial intelligence approaches is currently being investigated. As for pre-digitization curation workflows and recommendations, Nelson et al. [30] have provided a list of curatorial tasks for biological and palaeontological collections, while De Smedt et al. [20] have provided a useful checklist for pre-digitization curation steps, as a contribution to the DiSSCo Digitisation Guides website [31]. As for the automated extraction of metadata, there have been several studies [32,33,34,35]. Hedrick et al. [7] argued that the scientific community is ready for deeper exploration of minimal metadata capture using automation. Some approaches that use Optical Character Recognition (OCR), such as the SALIX Method [34], have been exploited at the Arizona State University Herbarium. More recently, the use of artificial intelligence algorithms has been proposed, to speed up metadata extraction [18,36]. An interesting workflow in this direction was recently described by Johaadien and Torma [37].

## 3. Digitization Workflows for Historical NHCs

The development of a digitization workflow focusing on historical NHCs (i.e., NHCs that were developed before 1960, usually following preparation criteria that were different from modern times) faces several challenges. In particular, it should minimize specimens’ handling, thus decreasing the risk of deterioration, while maximizing the efficiency of metadata extraction. Nelson et al. [8] outlined five major clusters of tasks in a digitization workflow: pre-digitization curation, image capture, image processing, data extraction, and geo-referencing. An effective digitization workflow should also be re-usable, and replicable, while most of the workflows in use are normally influenced by the context in which they have been developed [38].

Especially in the case of historical NHCs, which can barely be digitized by means of an industrial approach, there is a need for a modular, replicable, and scalable general workflow that takes into account issues such as specimen fragility and diversity, varying completeness of metadata, and the need for specialized curation and restoration (since historical NHCs often differ from modern NHCs because of changes in practices, technology, and purpose of curation). For example, digitization issues can arise from unmounted specimens, quite common in historical herbaria, which are not suitable for mass digitization efforts, since the process would be very slow and likely to damage the specimens. According to Guiraud et al. [39], such issues can be solved by decoupling the mounting and restoration of specimens from imaging, to lower the risk of a bottleneck in the workflow. It must also be noted that metadata extraction from historical specimens can pose significant challenges. While in modern collections metadata are normally present on labels only, in historical collections annotations may be present elsewhere. Furthermore, metadata may be present on manuscripts or documents associated with the specimens, or stored in separate archives. Lastly, historical NHCs often contain type specimens or other specimens of historical relevance that are not properly marked or exposed. Additionally, due to their dual role as both natural and cultural heritage, historical collections often call for specialized curatorial activities that are in between the approaches adopted for scientific collections and for “classic” cultural heritage.

## 4. A Case Study: The Algarium of Aristocle Vatova and Victor Schiffner

The algarium of Aristocle Vatova and Victor Schiffner, preserved at the Natural History Museum of Venice Giancarlo Ligabue, includes 1406 sheets with 2209 dried specimens, which were collected in 68 sampling sites in the Venice Lagoon between 1930 and 1932. The collection includes ca. 70 type specimens among the species, varieties, and forms. Despite its undeniable scientific and historical relevance, the collection has not been studied in the last 90 years, and is cited in the literature as “unknown” [40,41]. Each herbarium sheet hosts one or several specimens. The labels, written by various authors, usually detail taxon names and gathering localities. However, some taxon names are often noted on the herbarium sheet instead of the label. For instance, Figure 1 depicts multiple taxa listed on the sheet by Victor Schiffner. Additionally, sheets may include other annotations made by different authors. Microscopic slides are also present on several sheets, in protective paper envelopes, with their own annotations (Figure 1, top-right corner of the sheet).

The collection, which was used for a scientific report published in 1938 [42], is a snapshot of the algal community of the Venetian Lagoon between the two World Wars. Thus, it is of pivotal importance for understanding the temporal and spatial changes of macroalgal communities in the Lagoon over the last century. These changes are especially significant, given the exponential increase in human impact, notably from the development of the industrial area in Marghera, as well as from the digging of deep and large commercial canals between 1960 and 1970.

The digitization workflow was structured into six main steps:pre-digitization planning;pre-digitization curation;digital imaging;metadata transcription;geo-referencing;publication and dissemination.

### 4.1. Pre-Digitization Planning

The pre-digitization planning is extremely relevant in any digitization effort. Taking correct decisions in this phase leads to a smoother process, while poor choices can lead to wasted time and money, or even to the failure of the whole digitization.

In general, the pre-digitization planning should focus on checking whether personnel and adequate equipment are in place. Furthermore, it should take into account all those issues which could potentially bias the digitization effort, thus providing a reliable recovery plan. Given that funding is often one of the major limits [43], particular attention should be given to cost estimation. Examples of how detailed analysis for cost estimation should be carried out can be found in Hardisty et al. [44] and Walton et al. [45]. Assuming that cost analysis has been properly carried out, other questions should be answered while planning the digitization effort, especially if the target is a historical NHC:*Will a unique identifier be given to each specimen?* Unique Identifiers (UIDs) are particularly relevant, since they allow us to address univocally each specimen in a collection. Normally, they are associated with the specimen as a written alphanumeric string, a barcode, or a QR-code. Meadows et al. [46], Juty et al. [47], Hardisty et al. [48], and Islam S. et al. [49] have covered well the topic of Persistent Identifiers (PIDs) to properly identify digital representations of physical specimens, suggesting the assigning of a unique identifier to every locality and collection date, to properly track specimens. According to Hardisty et al. [48], preferred identifiers for NHCs specimens include CETAF Stable Identifiers [50], International Geo Sample Numbers (IGSNs) [51], GUIDs, Darwin Core Triplets, institution/collection codes, and catalog numbers.*Will full imaging be performed?* The specimens may have been preserved in envelopes, boxes, tubes, or other containers. Especially when envelopes have been used, labels or annotations have often been placed on the external surface of the envelope, while the specimen(s) are not visible unless the envelope is opened. This can be true also for specimen parts, microscopic slides, and other ancillary material linked to a specimen. In such cases, full digital imaging would require the opening of each envelope and the image label(s) and/or annotations being together with the specimen(s), thus achieving a complete digital representation. Alternatively, it is possible to focus on imaging the label(s) alone, greatly speeding up the process, with obvious impact on the duration and the cost of the digitization process.*Which image resolution, quality of image, and format will be adopted?* The International Image Interoperability Framework (IIIF, [52]) provides recommendations for digitizing cultural heritage materials, emphasizing the need for high-resolution images for detailed examination and research. As noted by Nieva de la Hidalga et al. [18], the majority of botanical institutions follow the digitization guidelines of the Global Plants Initiative (GPI) [53], which specifies the elements to include and the resolution for herbarium sheet images. According to Takano et al. [54], images should be usable and suitable for long-term storage. Capturing and preserving high-quality specimen images offers opportunities to take advantage of future improvements in image analysis, Optical Character Recognition (OCR) [55], natural language processing, handwriting analysis, and data-mining technologies [30]. While the minimum resolution for digital images should be set at 300 dpi, a resolution of 600 dpi is recommended, to capture fine details. However, image resolution should be a trade-off between quality and equipment costs, which can be relevant. Given that digital images are not included in the Minimum Information about a Digital Specimen (MIDS) up to level 2 [56,57], a lower resolution can be acceptable in the case of equipment restrictions. A trade-off should also be decided between quality and storage, given that permanent digital storage for images can be quite expensive. While a TIFF format allows for higher quality, it also calls for larger digital storage. On the other hand, JPEG images require far less (approx. 10^2^) storage room, but are of lower quality than other formats. Another option is the adoption of PNG compression. The latter is a *lossless* compression method, while the JPEG is a *lossy* one. Thus, the quality of PNG images is intrinsically higher. This, however, comes at the price of a larger file size, even if not comparable with that of TIFF files.*Which workflow type will be adopted?* Nelson et al. [30] described three dominant digitization workflows for natural history collections: (a) data capture with occasional specimen imaging, (b) parallel data and specimen imaging, and (c) imaging of specimens and labels followed by data capture from the images (*image-to-data* workflow). The latter has multiple advantages. It requires handling the specimens only once (for the digital imaging phase), while the transcription is made from the images. Furthermore, it allows for different operators to work on separate tasks (digital imaging and transcriptions), thus significantly increasing throughput [36]. In general, an *image-to-data* workflow should always be preferred, since it at least allows for reduction of the times a specimen is handled during the process (ideally down to once) if digital imaging is planned.*Will all the annotations on a specimen be transcribed?* The Minimum Information about a Digital Specimen (MIDS) [56,57] framework provides a structured approach to the digitization of natural history specimens. Each MIDS level provides a certain amount of information, which increases from level 0 to level 3. As an example, MIDS level 1 includes basic metadata, such as taxon name and gathering locality, while higher levels incorporate more detailed information [22]. Labels are the primary source of information about a specimen, especially as far as taxon name, locality, and date of collection are concerned. However, other annotations can be present on the label or elsewhere (on the herbarium sheet, on other labels, etc.). Annotations are not normally the focus of digitization [30], but they can be fundamental to the fitness-for-use of specimens [58]. While transcribing at least MIDS levels 1–2 generates a dataset fit for most uses, all the remaining annotations, which can be quite rich, can be of great interest as well, and may broaden the array of the potential uses of the specimens. However, their extraction may require a large investment, in terms of time and money. Again, a trade-off between costs and results should be achieved.*Which data scheme will be adopted?* The choice on how the data will be organized is closely related to the previous point, i.e., what will be transcribed. As an example, if the transcription focuses on MIDS up to level 3 then the Simple Darwin Core [59] flat structure can be easily adopted for organizing the data in a widely adopted standard format. However, if the transcription results in more complex data structures, with relationships different from 1:1, the Simple Darwin Core flat structure may be more difficult to adopt. For instance, if a specimen undergoes one or more revision event, the relationship between the specimen and the identification events will be 1:many. Thus, a proper representation of the specimen in the digital domain would call for a relational structure, instead of the flat structure of Simple Darwin Core [59]. The Darwin Core Archive overcomes this issue, allowing for 1:many relationships, thanks to the use of extensions, such as *Identification History*, *Measurement or Fact*, or *Simple Multimedia*. These extensions allow for relating one or more events or images to a single observation/specimen. At the same time, the ABCD standard [60], which was specifically designed to represent specimens and their 1:many relationships with events, data, and media, may be more useful in this case. In general, once a decision has been made on the amount and type of data that will be extracted from the specimens, a data model should be selected, taking into account also that data should be made interoperable on digital platforms, such as the Global Biodiversity Information Facility (GBIF) [61]. Since the Darwin Core Archive can accommodate specimens data, and it is the data standard adopted in the GBIF (but not only in the GBIF), its adoption should be preferred.

Before the digitization of the algarium of Aristocle Vatova and Victor Schiffner, the following decisions were made:Given that one-to-many specimens were present on each sheet, and given the structure of the digital archive of the Museum, which allowed for a single code for each herbarium sheet, it was decided to give to each specimen a UID, which would be made by an alphanumeric string composed by the code of the sheet followed by a sequential number for the specimen (e.g., MSNVE-0001234.1). When the dataset was published in the GBIF, CETAF stable identifiers would be adopted as well.During the digital imaging phase, all the envelopes were to be opened, and all the microscopic slides were to be digitized. This would call for a 1:many relationship between the specimens and the images, i.e., for each specimen, one or more images would exist.Images of the whole herbarium sheets were to be taken with a planetary scanner, while close-up images of particular morphological features, as well as for some slides, were to be captured, using a full-frame reflex camera. The images were to be stored as JPEG compressed files. The adoption of JPEG compression was due to the necessity of limiting the storage requirements, even in comparison with PNG compressed files, while ensuring an acceptable level of image quality to the viewers. Whenever possible, however, the adoption of higher quality formats is suggested. Image data were to be organized following the terms of the *Simple Multimedia extension* of the Darwin Core Archive in the version of the dataset that was to be published in the GBIF.An *image-to-data*-(to web) workflow was to be adopted. Specimens were to be digitally imaged, and then stored again, while transcription was to be performed using the digital images.Primary data and annotations were to be transcribed. Notes on each specimen about peculiar annotations or interventions were to be reported in the dataset. Annotations in other languages (i.e., German manuscript by V. Schiffner) were to be exposed to the users by means of digital images as extensions of the specimens [14].The Darwin Core standard model was to be adopted. In particular, among the Darwin Core terms, the following were to be used: catalogNumber; verbatimIdentification; scientificName; verbatimLocality; locationID; decimalLatitude; decimalLongitude; eventDate; recordedBy; recordedByID; identifiedBy; identifiedByID; identificationRemarks; occurenceRemarks. Furthermore, a non-standard term (Notes) was to be added for reporting annotations of scientific and historical interest that arose during the digitization process. This term was to be organized following the terms of the *Measurements or Facts* extension of the Darwin Core Archive in the version of the dataset that was to be published in the GBIF.

### 4.2. Pre-Digitization Curation

The actual workflow began with a pre-digitization curation phase. During this stage, the specimens were organized, catalog numbers were applied, and any necessary conservation or restoration work was performed, including the replacement of rusty pins, the fixing of loose specimens, and the relocation of labels that were loose or fixed to an incorrect sheet. In this phase, an investigation was also carried out, which was aimed at better understanding the collection’s historical context and the role of its authors.

### 4.3. Digital Imaging

During this phase, panoramic and close-up images of the specimens and microscopic slides were acquired. This allowed for the production of a repository of digital images, which were organized by specimen ID. In addition, all manuscripts associated with specimens were imaged as well. Envelopes containing slides were opened, and their labels were imaged together with the slides. If labels or notes glued to the sheet were covering the specimen then they were turned over and an additional image was captured. As a result, multiple images per specimen were produced (relationship many:1), while, when multiple specimens were present on a sheet, each image was also associated with many specimens (relationship 1:many). The same happened for the manuscripts, since they normally referred to more than one specimen, often on different herbarium sheets.

The equipment for digital imaging of the panoramic views was an entry-level planetary scanner (CZUR M3000 Pro Book Scanner, with a resolution of 16 megapixels), whereas close-up images were taken using a Canon 600D full-frame sensor (18 megapixels) with tethered remote shutter release, equipped with a 17–50 mm Tamron lens and a macro 105 mm Sigma. The light source for the camera was a custom-built copy stand Kaiser Repro RS 1 with 1 m column, 45 × 50 cm base with RB 218N HF led lighting. A scale bar and color standard (ColorChecker Calibrite classic Nano) were applied to each specimen before imaging.

### 4.4. Metadata Transcription

The labels and notes, mostly written by Aristocle Vatova, Victor Schiffner, Achille Forti, and other agents (also during a later rearrangement of the collection) were all handwritten. Often, barely readable annotations made with a pencil were present on the labels or on the sheets. In general, there was significant diversity in the writing styles and languages (mostly Italian and German, but often Latin as well). Several manuscripts, all produced by Victor Schiffner, mostly hosted diagnoses of newly described taxa in Latin and German. Metadata transcription was carried out manually. This required an expert assessment involving the understanding of label information, identifying targeted data elements, and selecting the appropriate element when multiple similar elements were present, e.g., sequences of identifications by different authors, such as Vatova and/or Forti followed by Schiffner.

### 4.5. Geo-Referencing

The sampling sites were geo-referenced into numerical coordinates (DwC terms: decimalLatitude, decimalLongitude). Verbatim locality data were transcribed, and the localities were associated with location IDs provided by the authors [42]. The geo-referencing process followed the Guide to Best Practices for Geo-referencing [62]. The geo-referencing relied on the centroid of the sampling site, as depicted on a map published by the authors [42]. Uncertainty, which is crucial for determining a data record’s fitness for use [63], was assessed on the basis of the radius of the point on the original map. The uncertainty was then fine-tuned for each sampling site by means of an expert assessment based on the geomorphology of the lagoon at the time of the original collection.

### 4.6. Publication and Dissemination

Metadata and images of the collection were made available online [64]. There exist several resources that expose algal specimens online, other than the GBIF. Among them, it is worth mentioning the resources of single institutions, such as the portal to the algae collections of the Natural History Museum of London [65] and that of the algae collection of the Muséum national d’Histoire naturelle of Paris [66], as well as multi-center resources. Among the latter, there exist thematic resources, such as the Macroalgal Herbarium Consortium [67], or more generalist resources, such as the Finnish Biodiversity Information Facility [68] and the JACQ [69], among several others. All these resources are mostly focused on addressing the community of researchers. The portal that was developed to publish the data deriving from the digitization of the Vatova–Schiffner collection was also developed to be useful to experts. However, it also provides several resources, such as insights on the authors and information on the history of the collection and on the Lagoon, and it showcases the specimens in a way that could be of interest to laypersons as well.

In the portal, the taxon names are those originally adopted by the authors in the monograph [42], and, to date, they have not been updated to modern nomenclature. In general, synonymization is rarely straightforward, and in most cases it requires a thorough taxonomic investigation by an expert taxonomist, especially when working on historical material. In the case of this collection, it is often quite complicated to refer the taxonomic concepts of Schiffner to modern taxonomic delimitations. In the portal, each specimen is linked to label metadata and the geo-referenced sampling sites in which it was collected. After a further review and quality control of the data (in progress), they will be published as a dataset in the GBIF network.

The digitization and its results have been showcased to the general public by means of press releases and public conferences, and to a scientific audience at the conferences of the Italian Association of Natural History Museums, and at the IX International Plant Conference, organized by the Italian Botanical Society.

## 5. Discussion

This contribution aims at offering an overview of the planning and execution of a digitization project carried out on a historical collection. The digitization was aimed not only at mobilizing scientific data, but also at exploiting the historical and cultural relevance of the collection. This contribution focuses mainly on digitization planning, providing a general guideline for planning and carrying out similar efforts.

The mobilization of specimens and specimen data through digitization has an obvious positive effect on NHMs and on the public. Mobilization widens the potential number of users of NHCs and specimens, and it greatly increases their fitness for use, by coupling specimen data with other scientific data, as well as historical information as extensions of the specimens [15], as also emphasized by Berents et al. [70]. Thanks to digitization and mobilization, NHMs can better showcase their collections, potentially attracting more visitors. Plus, digitizing their collections allows them to discover what they actually have in their warehouses. In fact, often a collection is a sort of “black box”, whose content is mostly unknown, and this is often especially true for historical collections (see, as an example, [71]).

Most historical collections are not suitable for industrial approaches to digitization [17,26]. These approaches emphasize automation: they are cost-effective when adopted for large, modern collections; however, historical collections normally host relatively limited amounts of specimens, and they require particular care in the handling and curating of the specimens before, during, and after the digital imaging. In addition, in historical NHCs specimens often have a certain diversity of size, shape, and preservation method. Furthermore, labels are mostly handwritten and diverse, in terms of language, size, content, and number, calling for interpretation during transcription.

In this case study, the data were manually transcribed from images. This approach, while time-consuming, is particularly suited to historical collections, whose specimens often host handwritten and unorganized data. Although computer vision and machine learning approaches, as suggested by Hedrick et al. [7], have shown promising results for handwriting recognition systems applicable to automated label transcription, their performance is optimal with printed text, or at least with well-organized handwritten texts, while they still yield mixed results on less organized handwritten labels [19].

The organization of metadata of digital specimens can pose some challenges. Data are often organized in spreadsheets, in a flat file format, i.e., in structures which allow for 1:1 relationships alone [72], even if this common practice produces several drawbacks (see, as an example, [73]). Data organized in spreadsheets can be easily accommodated into the Simple Darwin Core [59]. However, a complete representation of a digital specimen may require more complex relationships, due to, for example, the presence of multiple images per specimen or of sequences of events. As an example of the latter, in the algarium of Vatova and Schiffner, the labels report the activity of several agents, the first of whom was normally Aristocle Vatova, who often reported field notes and identifications. A further agent was Achille Forti, an algologist from Verona, who identified a large part of the material before declining to contribute further to the collection. A third agent was Victor Schiffner, who reviewed all the material originally identified by Forti, and who identified all the remaining specimens. Then, a fourth agent rewrote many of the labels, overwriting existing information (often making evident mistakes). Other than these agents, notes from other researchers are present. Understanding and highlighting the sequence of events by different agents was relevant to understanding the history of the specimens and the collection. However, their representation required a relational model, in which a single specimen was linked to many events. The extensions of Darwin Core could have been adopted, to organize these more complex relationships. However, in the case of the algarium, it was decided to accommodate the multiple identification events into the Darwin Core concept, “identificationRemarks”, describing the sequence of events as a text, instead of using the Darwin Core Archive extension *Identification history*, which was specifically tailored for the goal. This solution was far from ideal, but it was the simplest and least time-consuming. Furthermore, this solution was also adopted because the identification events often lacked a date or other relevant information. As a consequence, their agents, sequence, and results could sometimes only be hypothesized, and were probably better expressed as textual annotations. However, during the revision of the dataset for its publication in the GBIF, the adoption of the *Identification history* extension will be taken into account whenever possible. On the contrary, it was decided to adopt the Simple Multimedia extension for images of the specimens (there could be more than one image per specimen) and of the manuscripts, which reported the diagnoses of newly described taxa (by Victor Schiffner), and which were normally associated with many specimens.

The experience described in this contribution can be generalized to most historical collections, and could be used as a guideline for their digitization and mobilization. While time-consuming, the digitization of historical collections, especially if it also takes into account historical information, can be of pivotal significance for increasing their fitness for use not only in research but also as cultural heritage. NHMs, especially, could greatly benefit from the digitization of historical NHCs, making them their flagships, and building around them novel storytelling for physical and virtual exhibits.

## Figures and Tables

**Figure 1 plants-13-02901-f001:**
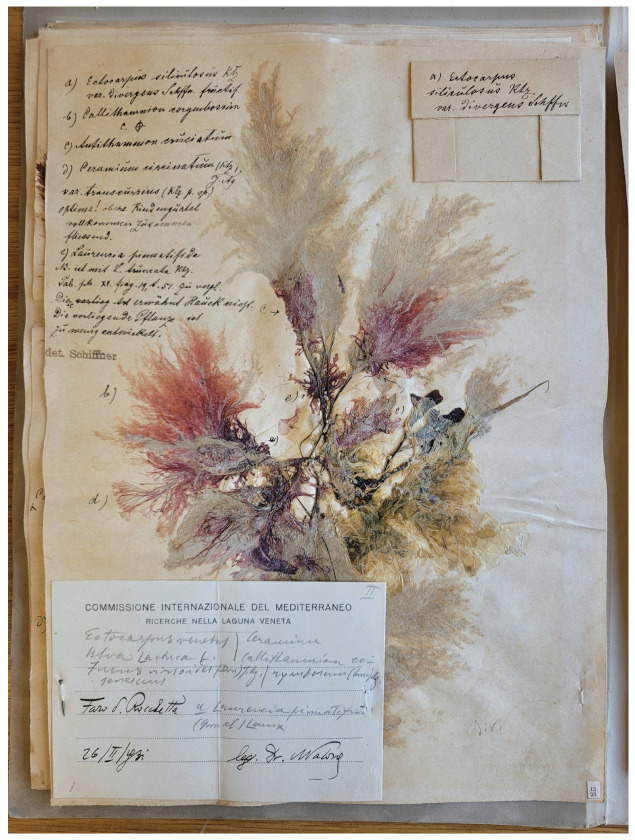
An example of a herbarium sheet from the algarium of Aristocle Vatova and Victor Schiffner. Note that taxon names, as well as other information, are present both on the label and on the sheet. An envelope containing a microscopic slide is also present at the top-right corner of the sheet.

## Data Availability

The original contributions presented in the study are included in the article, further inquiries can be directed to the corresponding author.

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
