# Peer review of "Valorization of Historical Natural History Collections Through Digitization: The Algarium Vatova–Schiffner"

_plants, 2024, doi:10.3390/plants13202901_

Round 1
Reviewer 1 Report
Comments and Suggestions for Authors
Valorization of historical natural history collections through digitization: the algarium Vatova-Schiffner
By L. Seggi, R. Trabucco & S. Martellos
The paper describes the workflow of the digitization of a relatively small macro algae collection from Venice. The introduction (chapters 1-3) gives a good and fully referenced overview for readers not experienced in herbarium digitization.
Chapter 4 documents the workflow for the concrete digitization project.
Chapter 5, the discussion, expands the workflow with some experiences and some remarks on the value of digital collections.
The manuscript has some merits, but is not very innovative. It might be sharpened in some details because some of the aspects have not been thought through with respect to data reusage and potential added value through scientific usage of the digitized data.
First of all, it is a pity, that the data collection is not yet available at GBIF. There is no proof that the mapping functions. The database, available in Italian language only, can be used by specialists. But as the label data are digitized as they have been written, a larger part of the data is not standardized according to the community standards. And the digitization of the handwritten sign on top of a “u” (e.g. AcrochaetiÅm with two unversioned scans) just to differenciate it from a hand-written “n” makes scientific name strings may be unlinkable for data portals using checklists as backbones. Another point is, that it is complicated to find the herbarium in data bases. The main identifier for herbaria is the Index herbariorum. It seems that the herbarium “Natural History Museum of Venice Giancarlo Ligabue” (lines 124-125) is a newer name for “Museo di Storia Naturale di Venezia” = herbarium Code MCVE with the second author as curator. This is very easy for the authors to explain, but complicated for people not familiar with Italian herbaria to study. May be the dataset metadata can be registered at GBIF and referenced in the paper. On the other hand, the beginning of a sample specimen is given as “MSNVE”. This is not the expected Herbarium Code nor any herbarium code known in Index Herbariorum.
Line 200: JPEG images cannot be recommended. Even though not used by the project PNG should be mentioned here. And today storage should not be a main problem, particularly TIFFs can be stored on separate drives as back-up.
Line 247: “Since GBIF … Darwin Core, its adoption as a standard should be preferred.” This might be an explanation why this is used but it is not a good argument why it “should” be used. The question is, if Darwin Core has all fields/features that are needed.
Line 252/254: UID’s are necessary (that’s trivial), but on an international level UUID’s or Stable identifiers are more appropriate. Why you do not implement the CETAF stable identifiers, you cited above? Or at least write that you will implement this, if not already done.
Line 260: again, JPEG cannot be recommended.
Line 289: Another point is, that you have different object that seem to be stored under one UID (herbarium sheet, preparation on glass, label, annotation sheets), this might be a problem.
Lines 330-333: Taxon names are those originally adopted by the authors in the monograph [40]. They were not updated, since even if some synonymization can be straightforward, in many cases a thorough taxonomic investigation by an expert taxonomist is necessary to refer the taxonomic concepts of Schiffner to modern taxonomic delimitations.
While the first sentence can stay as is, the reviewer has two comments on the second sentence which should be part of the discussion, (i) synonymization is seldom straightforward, because each specimen has to revised by experts. (ii) How this revision should be performed. Should an expert look up the images on the Web, write an anntotaion sheet, send it to you and you print this and make a new scan and add the metadata? Why you do not implement a digital annotation system on your web site (e.g. AnnoSys, see: Tschöpe et al. 2013 in Taxon 62(6): 1248-1258; Suhrbier et al. 2017 in Database 2017: bax018). This could solve workflow problems after digitization with state of the art 1930s. It is worth to discuss this problem, not tackled in the ms.
References:
The authors might consider to cite: De Smedt et al. (2024) in Phytokeys 244.
Author Response
Comments 1: The paper describes the workflow of the digitization of a relatively small macro algae collection from Venice. The introduction (chapters 1-3) gives a good and fully referenced overview for readers not experienced in herbarium digitization.
Chapter 4 documents the workflow for the concrete digitization project.
Chapter 5, the discussion, expands the workflow with some experiences and some remarks on the value of digital collections.
The manuscript has some merits, but is not very innovative. It might be sharpened in some details because some of the aspects have not been thought through with respect to data reusage and potential added value through scientific usage of the digitized data.
Response 1: we are grateful to the reviewer for his suggestions. We are providing an answer to all the comments here below.
Best regards
On behalf of the authors,
Stefano Martellos
Comments 2: First of all, it is a pity, that the data collection is not yet available at GBIF. There is no proof that the mapping functions.
Response 2: As we stated in the manuscript, the publication in the GBIF will be done as soon as the issues related to the alignment of names used by Schiffner to modern nomenclature will be solved. Our plan is to have it publishes as a data paper by mid 2025.
Comments 3: The database, available in Italian language only, can be used by specialists.
Response 3: At the moment the portal and the database are in Italian, but their translation in English will be carried out, hopefully by the end of the year.
Comments 4: But as the label data are digitized as they have been written, a larger part of the data is not standardized according to the community standards. And the digitization of the handwritten sign on top of a “u” (e.g. AcrochaetiÅm with two unversioned scans) just to differenciate it from a hand-written “n” makes scientific name strings may be unlinkable for data portals using checklists as backbones.
Response 4: to us it was extremely relevant to transcribe the labels verbatim, without performing any inference. This is the reason why we also are reviewing the dataset carefully before publishing it in the GBIF, which is in our plans since the very beginning. In fact, the Museum is about to start the process of becoming a publisher in the GBIF, and plans to publish not only this dataset, but other datasets as well, produced by digitization of other historical collections. We will make sure that the dataset will be fully compliant for data portals which make use of checklists as backbones.
Comments 5: Another point is, that it is complicated to find the herbarium in data bases. The main identifier for herbaria is the Index herbariorum. It seems that the herbarium “Natural History Museum of Venice Giancarlo Ligabue” (lines 124-125) is a newer name for “Museo di Storia Naturale di Venezia” = herbarium Code MCVE with the second author as curator. This is very easy for the authors to explain, but complicated for people not familiar with Italian herbaria to study.
Response 5: we agree that things are quite complicated here. The registration in Index Herbariorum is not only outdated, but also incorrect. In fact, it does not list all the plant collections of the Museum, but a generic “Herbarium”, wrongly stating that it is a fungarium only. In fact, the number of specimens listed in Index Herbariorum is approximately that of the actual fungarium, but the general herbarium itself is far richer, and contains several collections of vascular plants, algae, lichens, mosses and liverworts, other than macrobasidiomycetes. We noticed this recently, during a parallel activity to review the data of Italian institutions and collections in GrSciColl, which we are carrying out in collaboration with the Italian Association of Natural History Museums. Since GrSciColl harvested metadata from Index Herbariorum, we noticed the issue as soon as we started to review the metadata. The personnel of the Museum will update the information on Index Herbariorum asap.
Comments 6: May be the dataset metadata can be registered at GBIF and referenced in the paper.
Response 6: as in a previous answer, the Museum is starting the process to become a publisher in the GBIF, and then it will start to publish datasets, among which this one. The process is slightly slow, since the Museum is not an independent entity, but has to have green light by other administrative entities. However, it seems that the request for becoming publisher will be issues this month.
Comments 7: On the other hand, the beginning of a sample specimen is given as “MSNVE”. This is not the expected Herbarium Code nor any herbarium code known in Index Herbariorum.
Response 7: This is the code the Museum is using for cataloging all of its specimens (animals and plants). It is adopted since the very beginning, and cannot be changed.
Comments 8: Line 200: JPEG images cannot be recommended. Even though not used by the project PNG should be mentioned here. And today storage should not be a main problem, particularly TIFFs can be stored on separate drives as back-up.
Response 8: we added a line, stressing the benefits of using PNG as well. However, we would like to stress that storage can be a problem, depending on the budget an institution has. We do not know well the situation in other countries, but in Italy it is estimated that there exist ca. 450 institutions which host natural history collections. Most of them ha little, if no funding at all. Plus, we lack a National strategy for digitization and preservation of digitized specimens. Thus, each institution digitize and preserve digital specimens on its own budget. Renting storage, even if it could be seen as a minor expense, could anyway be a relevant issue. Plus, often Museums have no technical departments to carry out any part of the digitization process, including data and image storage and publication.
Furthermore, it is worth mentioning a discussion which was held at a plenary meeting of the recently ended COST Action MOBILISE, one of the projects aiming at building DiSSCo. The discussion topic was: “Should we delete images used in image-to-data workflows after transcribing the labels, since storage is expensive?”. Personally, we are quite disturbed by the idea of not having at least a JPG image for each digitized specimen. Better resolution files could be preferable, when possible due storage limitations.
The added line states that: “Another option is the adoption of the PNG compression. The latter is a loseless compression method, while the JPEG is a lossy one. Thus, the quality of PNG images is intrinsically higher. This however comes with the price of a larger file size, even if not comparable with that of TIFF files.”
Comments 9: Line 247: “Since GBIF … Darwin Core, its adoption as a standard should be preferred.” This might be an explanation why this is used but it is not a good argument why it “should” be used. The question is, if Darwin Core has all fields/features that are needed.
Response 9: Well, the answer to this question is complicated. The main advantage of ABCD is that it was developed to accomodate metadata of specimens. This is not true for Darwin Core, and this is the reason so many extensions were developed to convert the simple Darwin Core into the Darwin Core Archive, and the effort that is being carried out for re-designing the Darwin Core model.
Anyway, we agree that the line as we wrote is not a good argument, and we modified it as follows: “Since Darwin Core Archive can accomodate specimens data, and it is the data standard adopted in the GBIF (but not only), its adoption should be preferred.”
Comments 10: Line 252/254: UID’s are necessary (that’s trivial), but on an international level UUID’s or Stable identifiers are more appropriate. Why you do not implement the CETAF stable identifiers, you cited above? Or at least write that you will implement this, if not already done.
Response 10: we added to the manuscript a line stating that we will adopt CETAF stable identifiers in the dataset, once we will finalize it for publishing in the GBIF. The line is as follows: “When the dataset will be published in the GBIF, CETAF stable identifiers will be adopted as well.”
Comments 11: Line 260: again, JPEG cannot be recommended.
Response 11: We modified this line suggesting the use of PNG format as well.
Now the text is as follows: “Images were stored ad JPEG compressed files. The adoption of the JPEG compression was due to the necessity of limiting the storage requirements, even in comparison with PNG compressed files, while ensuring an acceptable level of image quality to the viewers. When possible, however, the adoption of higher quality formats is suggested.”
Comments 12: Line 289: Another point is, that you have different object that seem to be stored under one UID (herbarium sheet, preparation on glass, label, annotation sheets), this might be a problem.
Response 12: the UID we cited in the text refers to the specimen. All other entities which were created during the digitization process have their own UID in the database.
Comments 13: Lines 330-333: Taxon names are those originally adopted by the authors in the monograph [40]. They were not updated, since even if some synonymization can be straightforward, in many cases a thorough taxonomic investigation by an expert taxonomist is necessary to refer the taxonomic concepts of Schiffner to modern taxonomic delimitations.
While the first sentence can stay as is, the reviewer has two comments on the second sentence which should be part of the discussion, (i) synonymization is seldom straightforward, because each specimen has to revised by experts. (ii) How this revision should be performed. Should an expert look up the images on the Web, write an anntotaion sheet, send it to you and you print this and make a new scan and add the metadata? Why you do not implement a digital annotation system on your web site (e.g. AnnoSys, see: Tschöpe et al. 2013 in Taxon 62(6): 1248-1258; Suhrbier et al. 2017 in Database 2017: bax018). This could solve workflow problems after digitization with state of the art 1930s. It is worth to discuss this problem, not tackled in the ms.
Response 13: We agree that synonimization should often require the control of a taxonomist, and this is especially true for historical collections. We have recently digitized other two historical collections of lichens, and we are facing similar problems (thus the need for careful review before publishing the data). However, in some cases the process is almost straightforward.
On the how the revision will be performed, we are cooperating with algologists in the Universities of Catania and Trieste to carry out the process. This is the reason we are not implementing a digital annotation system (which, anyway, would be out of the scope of the present research). This is also the reason why we did not discuss the topic any further in the manuscript.
However, we modified the paragraph in lines 331-334 as follows: “Taxon names are those originally adopted by the authors in the monograph [40] and, at the moment, they were not updated to modern nomenclature. In general, synonymization is rarely straightforward, and in most cases it requires a thorough taxonomic investigation by an expert taxonomist, especially when working on historical material. In the case of this collection, it is often quite complicated to refer the taxonomic concepts of Schiffner to modern taxonomic delimitations.”
We hope that this change will satisfy the reviewer.
Comment 14: References: The authors might consider to cite: De Smedt et al. (2024) in Phytokeys 244.
Response 14: we agree that the reference is relevant, and we cited this paper at line 59.
Reviewer 2 Report
Comments and Suggestions for Authors
The article is interesting and innovative. the study project is well organized and carried out. The investigation methods are well illustrated. The conclusions are consistent with what was investigated. The proposed protocols can constitute an excellent path for studies and valorization of both historical and modern Herbaria.
In row112, the bibliographic reference [50] is out of order; it is advisable to re-establish the numerical sequence.
Author Response
Comments 1: The article is interesting and innovative. the study project is well organized and carried out. The investigation methods are well illustrated. The conclusions are consistent with what was investigated. The proposed protocols can constitute an excellent path for studies and valorization of both historical and modern Herbaria.
In row112, the bibliographic reference [50] is out of order; it is advisable to re-establish the numerical sequence.
Response 1: Thank you very much for your review.
We have re-established the numerical sequence of the references
Best regards
On behalf of the authors,
Stefano Martellos
Reviewer 3 Report
Comments and Suggestions for Authors
This manuscript is an excellent contribution to the digitization standards of algae worldwide, as well as to historical collections. I really enjoyed reading through the manuscript and how it was constructed. My only criticism is regarding the lack of discussion comparing their results with the digitization of other Algarium worldwide. If the authors are able to provide a short discussion on that matter, the manuscript will be in great shape for publication.
Author Response
Comments 1: This manuscript is an excellent contribution to the digitization standards of algae worldwide, as well as to historical collections. I really enjoyed reading through the manuscript and how it was constructed. My only criticism is regarding the lack of discussion comparing their results with the digitization of other Algarium worldwide. If the authors are able to provide a short discussion on that matter, the manuscript will be in great shape for publication.
Response 1: Thank you very much for your review.
Even if discussing the digitization of other Algae Herbaria is out of the scope of this manuscript, we added a short paragraph to the Results section, in the subsection 4.6. Publication and dissemination, highliting some of the mayor efforts in digitizing and publishing algal specimens metadata and media. We hope this could improve the manuscript enough.
The paragraph, which modifies and integrates the beginning of the subsection 4.6, is as follows: “Metadata and images of the collection were made available online. There exist several resources which expose algal specimens online, other than the GBIF. Among them, it is worth to mention resources of single institutions, such as the portal to the algae collections of the Natural History Museum of London and that to the algae collection of the Muséum national d’Histoire naturelle of Paris, as well as multi-center resources. Among the latter, there exist thematic resources, such as the Macroalgal Herbarium Consortium, or more generalist resources, which host algal specimens, but not only, such as the Finnish Biodiversity Information Facility and the JACQ, among several other initiatives. All these resources are mostly focused on addressing a community of researches.
The portal that was developed to publish the data deriving from the digitization of the Vatova-Schiffner collection was also developed to be useful to experts. However, it also provides several resources, such as insights on the authors and information on the history of the collection and on the Lagoon, and showcases the specimens in a way that could be of interest to laypersons as well.”
Best regards
On behalf of the authors,
Stefano Martellos